# New 1,2,4-Oxadiazole Nortopsentin Derivatives with Cytotoxic Activity

**DOI:** 10.3390/md17010035

**Published:** 2019-01-08

**Authors:** Stella Cascioferro, Alessandro Attanzio, Veronica Di Sarno, Simona Musella, Luisa Tesoriere, Girolamo Cirrincione, Patrizia Diana, Barbara Parrino

**Affiliations:** 1Department of Biological, Chemical and Pharmaceutical Sciences and Technologies (STEBICEF), University of Palermo, via Archirafi 32, 90123 Palermo, Italy; stellamaria.cascioferro@unipa.it (S.C.); alessandro.attanzio@unipa.it (A.A.); luisa.tesoriere@unipa.it (L.T.); girolamo.cirrincione@unipa.it (G.C.); patrizia.diana@unipa.it (P.D.); 2Department of Pharmacy, University of Salerno, Via G. Paolo II 132, 84084 Fisciano, Italy; vdisarno@unisa.it (V.D.S.); smusella@unisa.it (S.M.)

**Keywords:** marine alkaloids, nortopsentin analogs, 1,2,4-oxadiazole derivatives, anti-cancer agents, antiproliferative activity

## Abstract

New analogs of nortopsentin, a natural 2,4-bis(3′-indolyl)imidazole alkaloid, in which the central imidazole ring of the natural lead was replaced by a 1,2,4-oxadiazole moiety, and in which a 7-azaindole portion substituted the original indole moiety, were efficiently synthesized. Among all derivatives, prescreened against the HCT-116 colon rectal carcinoma cell line, the two most active compounds were selected and further investigated in different human tumor cells showing IC_50_ values in the micromolar and submicromolar range. Flow cytometric analysis of propidium iodide-stained MCF-7 cells demonstrated that both the active derivatives caused cell cycle arrest in the G0–G1 phase. The cell death mechanism induced by the compounds was considered to be apoptotic by measuring the exposure of phosphatidylserine to the outer membrane and observed morphological evaluation using acridine orange/ethidium bromide double staining. Moreover, further tested on intestinal normal-like differentiated Caco-2 cell line, they exhibited preferential toxicity towards cancer cells.

## 1. Introduction

Natural products (NP) constitute a significant source of bioactive molecules and potential drug leads due to their high chemical diversity, biochemical specificity, binding efficiency with biological targets, and broad panel of bioactivities. About 60% of drugs currently on the market are of natural origin and natural products screening still plays a fundamental role in the drug discovery process [1]. Marine natural products (MNP), in particular marine sponge-derived compounds, have attracted considerable attention due to their unique biodiversity and structural differences as compared to terrestrial natural products. Among MNP or marine-derived molecules, eight compounds are currently on the market with application in different therapeutic areas and several compounds are in different phases of the clinical pipeline, showing promising anticancer activity. Only a few of them are original MNPs, while the majority are derivatives obtained through molecular lead optimization [2].

Considering the successful results obtained using MNP as leads for drug discovery and the growing number of marine-derived molecules entering into clinical trials, researchers are still inspired by their scaffold for the design of new active molecules.

Among MNPs used as lead compounds for the synthesis of new anticancer agents, nortopsentin, an alkaloid isolated from deep-sea sponge *Spongsorites ruetzleri* having a characteristic 2,4-bis(3′-indolyl)imidazole skeleton, attracted remarkable attention due to its significant antiproliferative activity against the P388 murine leukemia cell line [3]. Many derivatives in which the central imidazole ring was replaced by several five-membered heterocycles were reported, most of them showing antiproliferative activity, often reaching GI_50_ values in the low micromolar range or even at the sub-micromolar level [3,4,5,6,7,8,9,10,11,12]. The thiazole nortopsentin analogs in particular, in which the structural modification of the lead natural molecule also involved one or both indole portions, led to potent compounds with effect against a wide range of cell lines, including diffuse malignant peritoneal mesothelioma (DMPM), a fatal disease, poorly responsive to conventional therapies, and acted as CDK1 inhibitors [13,14,15,16]. Moreover, the most active thiazole derivatives also bearing a 7-azaindole substitution, in the mouse model, by intraperitoneal administration were effective with a significant reduction of the DMPM and two complete responses at well-tolerated doses [13].

On the other hand, the 1,2,4-oxadiazole ring system is a five-membered heterocycle ring found in many molecules with significant biological activity, especially antitumor [17,18,19], despite its uncommon presence in MNP. To the best of our knowledge, phidianidines, isolated from the marine opisthobranch mollusk *Phidiana militaris*, are a unique example of MNP possessing a 1,2,4-oxadiazole ring system in their structure, and together with their derivatives they exhibit significant cytotoxic, DAT inhibitory, or neuroprotective activities [20,21,22]. Moreover, the 1,2,4 oxadiazole ring is a bioisoster of amides and esters and for this reason could improve bioavailability and physiochemical properties of compounds bearing it.

Continuing our search for new anticancer compounds [23,24,25,26,27,28,29,30,31,32,33,34,35,36,37,38,39,40], herein we report the synthesis of a new 1-methyl-3-[3-(1-methyl-1*H*-indol-3-yl)-1,2,4-oxadiazol-5-yl]-1*H*-pyrrolo[2,3-*b*]pyridine nortopsentin analog **1**, designed on the basis of the potent activity shown by thiazole nortopsentin analogs with a 7-azaindole portion and considering the important characteristics of the 1,2,4-oxadiazole ring found in compounds with promising biological activity. The new compounds, in which the central imidazole ring of the natural lead was replaced by a 1,2,4-oxadiazole moiety, and in which a 7-azaindole portion substituted the original indole, were prescreened against the HCT-116 colon rectal carcinoma cell line and the two most active compounds were selected and further investigated in different human tumor cell lines.

## 2. Results and Discussion

### 2.1. Chemistry

1-Methyl-3-[3-(1-methyl-1*H*-indol-3-yl)-1,2,4-oxadiazol-5-yl]-1*H*-pyrrolo[2,3-*b*]pyridines **1** were synthesized by reaction between two different key intermediates, *N′*-hydroxy-1-methyl-1*H*-indole-3-carboximidamides **2** and methyl-1-methyl-1*H*-pyrrolo[2,3-*b*]pyridine-3-carboxylates 3 (Scheme 1, Table 1).

*N′*-hydroxy-1-methyl-1*H*-indole-3-carboximidamides **2a**–**e**, commercially unavailable, were synthesized from the corresponding 1-methyl-indoles of type **4**, prepared as previously reported [16]. The reaction of compounds **4a**–**e** with chlorosulfonyl isocyanate (CSI) in acetonitrile, followed by the addition of *N*,*N*-dimethylformamide (DMF), led to the corresponding carbonitriles **5a**–**e** (90–98%), easily converted (65–82%) to the corresponding carboximidamides **2a**–**e** through reaction with hydroxylamine hydrochloride in ethanol (EtOH), in the presence of diisopripylenethylamide (DIPEA).

Methyl-1-methyl-1*H*-pyrrolo[2,3-*b*]pyridine-3-carboxylates **3a**–**c** were in turn prepared from their 1-methyl-7-azaindoles **6a–c** [10,16] converted into the corresponding 2,2,2-trichloro-1-(1-methyl-1*H*-pyrrolo[2,3-*b*]pyridin-3-yl)ethanones **7a**–**c** (80–99%) by reaction with trichloroacetyl chloride in dichloromethane (DCM) in the presence of aluminum chloride. Once obtained, compounds **7** were subjected to basic hydrolysis with potassium hydroxide solution (KOH 20%) to afford the desired derivatives **3** (90–99%).

Finally, the carboximidamides **2a**–**e** and the carboxylates **3a**–**c** were reacted in the presence of sodium hydride in tetrahydrofuran (THF) to afford the final compounds **1a**–**o** (55–85%).

### 2.2. Biology

In vitro cytotoxicity of the synthesized compounds **1a**–**o** was prescreened against the HCT-116 colon rectal carcinoma cell line. Monolayer cultures treated for 72 h with 10^−8^–10^−4^ µM concentrations of the compounds were examined by MTT assay for the cell viability.

Among the synthesized nortopsentin analogs, the compounds bearing the 5-bromo-1-methyl-1*H*-pyrrolo[2,3-*b*]pyridine moiety (see compounds **1k** and **1n**) showed the highest cytotoxic activity (Figure 1B). Substitution of the bromine atom in this portion with a fluorine (see compounds **1o** and **1l**) or absence of the halogen atom (see compounds **1j** and **1m**) resulted in a drop of the antiproliferative effect of the derivatives. The remaining compounds appeared poorly effective.

The most active derivatives **1k** and **1n** were further investigated against MCF-7 (human breast cancer), HeLa (cervix adenocarcinoma), and CaCo2 (colorectal carcinoma) cell lines (Figure 2), and IC_50_ values in the micromolar and submicromolar range were calculated (Table 2). Compared to **1n**, **1k** derivative, with the exception of HeLa cell line, appeared 2–4-fold more effective.

Further studies on the mechanism of the cytotoxic activity of the active compounds were carried out on MCF-7 cells, the tumor cell line more sensitive to both compounds. In order to investigate the effect on the cell cycle, flow cytometric analysis on PI stained MCF-7 cells was carried out after 24 h treatment with compounds **1k** and **1n** at their relevant IC_50_ values. As shown in Figure 3A, both the synthesized derivatives caused an accumulation of treated cells in subG0/G1 phase and induced their marked arrest (more than 60%) in G0–G1 phase of the cycle. Although changes in distribution within mitotic phases cannot provide precise information on the mechanism of drug’s activity, these findings suggest that compounds **1k** and **1n** can affect the cell machinery promoting DNA duplication.

In order to assess the cell death mechanism, apoptosis induction caused by compounds **1k** and **1n** in MCF7 cells, after 24 h treatment, was investigated by means of Annexin V/PI dual staining method followed by cytofluorimetric analysis. The biparametric analysis showed that both the compounds induced early apoptosis without causing death by necrosis (Figure 3B).

It is of interest to emphasize that other previously synthesized nortopsentin analogs bearing a 7-azaindole substitution and in which a thiazole ring substituted the central ring of the lead compound, showed cytotoxic activity towards MCF-7 cells associated with mitotic failure and accumulation of cells in phase G2/M [12,16]. The substitution of the central ring with a 1,2,4 oxadiazole one provides molecules of different biological activity that deserves to be investigated.

To further verify the pro-apoptotic effects of the synthesized derivatives, AO/EB staining was also used. Morphological evaluation of the cells using AO and EB double staining allows to detect early apoptotic cells stained green with yellow dots, with blebbing cytoplasm, while late apoptotic cells stain orange with fragmented nuclei. Non-apoptotic cells stain green. As shown in Figure 4, MCF-7 cells treated for 24 h with **1k** or **1n** showed morphological changes typical for early apoptosis with condensation of nuclear material and formation of membrane blebbing.

Additional experiments were conducted on intestinal normal-like differentiated Caco-2 cells. As shown in Figure 5, at the concentrations effective to inhibit the growth of tumor cells, the compounds **1k** and **1n** did not affect to a large extent the viability of the normal-like cells showing selectivity towards cancer cells (Figure 4).

## 3. Materials and Methods

### 3.1. Chemistry

#### 3.1.1. General

All melting points were taken on a Büchi-Tottoly capillary apparatus (Büchi, Cornaredo, Italy) and are uncorrected. IR spectra were determined in bromoform with a Shimadzu FT/IR 8400S spectrophotometer (Shimadzu Corporation, Milan, Italy). ^1^H and ^13^C NMR spectra were measured at 200 and 50.0 MHz, respectively, in DMSO-*d_6_* solution, using a Bruker Avance II series 200 MHz spectrometer (Bruker, Milan, Italy). Column chromatography was performed with Merck silica gel 230–400 mesh ASTM or with a Büchi Sepacor chromatography module (prepacked cartridge system). Elemental analyses (C, H, N) were within ± 0.4% of theoretical values and were performed with a VARIO EL III elemental analyzer (Elementar, Langenselbold, Germany). Purity of all the tested compounds was greater than 95%, determined by HPLC (Agilent 1100 Series). Mass spectra of final compounds were performed using a Mariner™ mass spectrometer, Applied Biosystems (Foster City, CA, USA). A Harvard model 11 syringe pump (Holliston, MA, USA) was used to infuse the sample solutions. The ESI source was operated in positive ion mode with an electrospray voltage of 4.5 kV. Compounds **1a**–**o** were characterized only by ^1^H NMR spectra due to their poor solubility.

##### General Procedure for the Synthesis of 1-methyl-1*H*-indole-3-carbonitriles (**5a–e**)

The appropriate indole **4a**–**e** (1.8 mmol) solubilized in anhydrous acetonitrile (2.0 mL), was reacted with chlorosulfonyl isocyanate (CSI) (0.16 mL, 1.8 mmol, added dropwise at 0 °C. The resulting reaction mixture was stirred at 0 °C for 2 h after which anhydrous dimethylformamide (DMF) (1.0 mL, 106.3 mmol) was added dropwise. The mixture was stirred at 0 °C for 1 h and then it was poured into ice-water. The obtained precipitate was filtered off, dried (Na_2_SO_4_), and purified by column chromatography using dichloromethane (DCM) as the eluent.

*1-Methyl-1H-indole-3-carbonitrile* (**5a**). White solid; yield 90% m.p. 61–62 °C; spectroscopic data are in accordance to those reported in literature [41].

*5-Bromo-1-methyl-1H-indole-3-carbonitrile* (**5b**). White solid; yield 92%; m.p. 103–104 °C; spectroscopic data are in accordance to those reported in literature [41].

*5-Fluoro-1-methyl-1H-indole-3-carbonitrile* (**5c**). White solid; yield 90%; m.p. 76.5–78 °C; spectroscopic data are in accordance to those reported in literature [41].

*5-Chloro-1-methyl-1H-indolo-3-carbonitrile* (**5d**). White solid; yield: 91%; m.p. 103.8–104.5 °C; spectroscopic data are in accordance to those reported in literature [42].

*5-Metoxy-1-metil-1H-indolo-3-carbonitrile* (**5e**). White solid; yield 98%; m.p. 106.5–107.4 °C; spectroscopic data are in accordance to those reported in literature [41].

##### General Procedure for the Synthesis of *N′*-hydroxy-1-methyl-1*H*-indole-3-carboximidamides (**2a–e**)

To a solution of the appropriate 1-methyl-1*H*-indolo-3-carbonitrile **5** (0.9 mmol) in ethanol (15 mL) diisopripylenethylamide (DIPEA, 1.32 mL) and hydroxylamine hydrochloride (2.3 mmol, 158.0 mg) were added and the resulting reaction mixture was heated under reflux for 4 h. The solvent was evaporated at reduced pressure and the obtained crude was suspended in water and extracted with ethyl acetate (3 × 30 mL). The organic phases were dried (Na_2_SO_4_) and evaporated under reduced pressure. The residue was purified by column chromatography using ethyl acetate as the eluent.

*N′-hydroxy-1-methyl-1H-indole-3-carboximidamide* (**2a**). solid; yield 65%; m.p. 110.2–111.6 °C; spectroscopic data are in accordance to those reported in literature [41].

*5-Bromo-N′-hydroxy-1-methyl-1H-indole-3-carboximidamide* (**2b**). White solid; yield 75%; m.p. 103.0–104.0 °C IR (cm^−1^): 3465 (OH), 3360 (NH_2_), 1669 (C=N); ^1^H NMR (200 MHz, DMSO-*d_6_*) δ: 3.79 (s, 3H, CH_3_), 5.64 (s, 2H, NH_2_), 7.30 (d, 1H, *J* = 8.7 Hz, H-6), 7.44 (d, *J* = 8.7 Hz, 1H, H-7), 7.79 (s, 1H, H-4), 8.27 (s, 1H, H-2), 9.34 (1H, s, OH); ^13^C NMR (50 MHz, DMSO-*d_6_*) δ: 32.9 (q), 107.8 (s), 111.9 (d), 112.4 (s), 124.0 (d), 124.4 (d), 126.4 (s), 130.0 (d), 135.6 (s), 148.6 (s); *Anal.* Calculated for C_10_H_10_BrN_3_O (MW: 268.11): C, 44.80; H, 3.76; N, 15.67%. Found: C, 45.11; H, 4.00; N, 15.48%.

*5-Fluoro-N′-hydroxy-1-methyl-1H-indole-3-carboximidamide* (**2c**). Gray solid; yield 80%; m.p. 114.0–115.0 °C; IR (cm^−1^): 3465 (OH), 3370 (NH_2_), 1640 (C=N); ^1^H NMR (200 MHz, DMSO-d_6_) δ: 3.79 (s, 3H, CH_3_), 5.62 (s, 2H, NH_2_), 6.98–7.09 (td, J = 9.2, 9.1, 2.5 Hz, 1H, H-6), 7.45 (dd, J = 9.1, 4.4 Hz, 1H, H-7), 7.74–7.80 (m, 2H, H-4 and H-2), 8.27 (s, 1H, OH); ^13^C NMR (50 MHz, DMSO-d_6_) δ:33.0 (q), 106.7 (d, J*_C4-F_* = 24.8 Hz), 108.5 (s, J*_C7a-F_* = 4.6 Hz), 109.7 (d, J*_C6-F_* = 26.1 Hz), 111.0 (d, J*_C7-F_* = 9.9 Hz), 125.0 (s, J*_C3a-F_* = 11.0 Hz), 130.3 (d), 133.6 (s), 148.8 (s), 157.3 (s, J*_C5-F_* = 231.9 Hz); *Anal*. Calculated for C_10_H_10_FN_3_O (MW: 207.20): C, 57.97; H, 4.86; N, 20.28%. Found: C, 57.88; H, 4.62; N, 20.31%.

*5-Chloro-N′-hydroxy-1-methyl-1H-indole-3-carboximidamide* (**2d**). Gray solid; yield 82%; m.p. 170.7–172.0 °C; IR (cm^−1^): 3461 (OH), 3364 (NH_2_), 1652 (C=N); ^1^H NMR (200 MHz, DMSO-d_6_) δ 3.79 (s, 3H, CH_3_), 5.66 (s, 2H, NH_2_), 7.19 (dd, J = 8.7, 2.2 Hz, 1H, H-6), 7.47 (d, J = 8.7 Hz, 1H, H-7), 7.81 (s, 1H, H-2), 8.13 (d, J = 2.2 Hz, 1H, H-4), 9.35 (s, 1H, OH). ^13^C NMR (50 MHz, DMSO-d_6_) δ 32.89 (q), 107.84 (s), 111.47 (d), 121.31 (d), 121.52 (d) 124.38 (s), 125.75 (s), 130.15 (d), 135.36 (s), 148.63 (s). *Anal.* Calculated for C_10_H_10_ClN_3_O (MW: 223.66): C, 53.70; H, 4.51; N, 18.79%. Found: C, 53.82; H 4.67; N, 19.00%.

*5-Methoxy-N′-hydroxy-1-methyl-1H-indole-3-carboximidamide* (**2e**). White solid; yield 75%; m.p. 165.9–166.5 °C; IR (cm^−1^): 3447 (OH), 3356 (NH_2_), 1652 (C=N); ^1^H NMR (200 MHz, DMSO-d_6_) δ 3.75 (s, 3H, CH_3_x2), 5.58 (s, 2H, NH_2_), 6.82 (dd, J = 8.9, 2.5 Hz, 1H, H-6), 7.33 (d, J = 8.9 Hz, 1H, H-7), 7.59 (d, J = 2.5 Hz, 1H, H-4), 7.69 (s, 1H, H-2), 9.28 (s, 1H, OH).^13^C NMR (50 MHz, DMSO-d_6_) δ 32.79 (q), 55.22 (q), 103.67 (d), 107.64 (s), 110.49 (d), 111.77 (d), 125.22 (s), 129.07 (d), 132.09 (s), 149.29 (s), 153.86 (s). *Anal.* Calculated for C_10_H_10_N_2_O (MW: 219.24): C, 60.26; H, 5.98; N, 19.17%. Found: C, 60.22; H, 5.67; N, 19.25%.

##### General Procedure for the Synthesis of 2,2,2-trichloro-1-(1-methyl-1*H*-pyrrolo[2,3-*b*]pyridin-3-yl)ethanones (**7a–c**)

To a solution of the appropriate 1-methyl-1*H*-pyrrolo[2,3-*b*]pyridine **6** (10.15 mmol) in DCM (24 mL), aluminum chloride (35 mmol, 4.7 g) and a solution of trichloroacetyl chloride in DCM (10.15 mmol, 1.1 mL in 5 mL di DCM), were added. The resulting mixture was stirred at room temperature for 6 h, poured into ice-water and extracted with DCM (3 × 20 mL). The organic phases were washed with HCl 10% (2 × 30 mL), dried (Na_2_SO_4_), and evaporated under reduced pressure, to afford the desired derivatives as pure compounds.

*2,2,2-Trichloro-1-(1-methyl-1H-pyrrolo[2,3-b]pyridin-3-yl)ethanone* (**7a**). White solid; yield 80%; m.p. 99–100 °C; IR cm^−1^: 1690 (CO);^1^H NMR (200 MHz, DMSO-d_6_) δ 4.00 (s, 3H, CH_3_). 7.42 (dd, J = 8.0, 4.8 Hz, 1H, H-5), 8.49 (dt, J = 4.8, 2.0 Hz, 1H, H-6), 8.54 (d, J = 2.0 Hz, 1H, H-4), 8.90 (s, 1H, H-2). ^13^C NMR (50 MHz, DMSO-d_6_) δ 31.95 (q), 95.84 (s), 102.37 (s), 119.40 (d), 119.82 (s), 129.97 (d), 140.09 (d), 144.90 (d), 147.64 (s), 176.30 (s). *Anal.* Calculated for C_10_H_7_Cl_3_N_2_O (MW: 277.53): C, 43.28; H, 2.54; N, 10.09%. Found: C, 43.32; H 2.67; N, 9.85%.

*2,2,2-Trichloro-1-(5-fluoro-1-methyl-1H-pyrrolo[2,3-b]pyridin-3-yl)ethanone* (**7b**). Yellow solid; yield 99%; m.p. 110–111 °C; IR cm^−1^: 1710 (CO); ^1^H NMR (200 MHz, DMSO-d_6_) δ 3.98 (s, 3H, CH_3_). 8.29 (dd, J = 9.0, 2.8 Hz, 1H, H-4), 8.51 (dd, J = 2.8, 1.7 Hz, 1H, H-6), 8.99 (s, 1H, H-2). ^13^C NMR (50 MHz, DMSO-d_6_) δ 32.31 (q), 99.49 (s), 120.43 (s), 102.29 (s), 115.74 (d, J*_C4-F_* = 29.3 Hz), 133.54 (d, J*_C6-F_* = 29.3 Hz), 141.80 (d), 140.22 (s), 144.40 (s), 160.20 (d, J*_C5-F_* = 70.4 Hz). *Anal.* Calculated for C_11_H_6_Cl_3_FN_2_O (MW: 295.52): C, 40.64; H, 2.05; N, 9.48%. Found: C, 40.55; H 1.98; N, 9.68%.

*2,2,2-Trichloro-1-(5-bromo-1-methyl-1H-pyrrolo[2,3-b]pyridin-3-yl)ethanone* (**7c**). Orange solid; yield 99%; m.p. 123–124 °C; IR cm^−1^: 1725 (CO);^1^H NMR (200 MHz, DMSO-d_6_) δ 3.97 (s, 3H, CH_3_), 8.57 (d, J = 2.2 Hz, 1H, H-4), 8.63 (d, J = 2.2 Hz, 1H, H-6), 8.96 (s, 1H, H-2).^13^C NMR (50 MHz, DMSO-d_6_) δ 32.21 (s), 101.97 (s), 115.00 (s), 121.41 (s), 131.69 (d), 141.40 (d), 145.23 (d), 146.24 (s), 176.32 (s). *Anal.* Calculated for C_10_H_6_BrCl_3_N_2_O (MW: 356.43): C, 33.70; H, 1.70; N, 7.86%. Found: C, 34.01; H 1.55; N, 7.68%.

##### General Procedure for the Synthesis of methyl-1-methyl-1*H*-pyrrolo[2,3-*b*]pyridine-3-carboxylate (**3a–c**)

To a mixture of the appropriate ethanone **7** (1.1 mmol) in methanol, KOH 20% solution (0.33 mL) was added. The reaction mixture was stirred at room temperature for 3 h. Then HCl 6N (0.2 mL) was added and the mixture was extracted in ethyl acetate (3 × 30 mL).

*Methyl-1-methyl-1H-pyrrolo[2,3-b]pyridine-3-carboxylate* (**3a**). White solid; yield 99%; m.p. 135–136 °C; IR cm^−1^: 1730 (CO); ^1^H NMR (200 MHz, DMSO-d_6_) δ 3.83 (s, 3H, CH_3_), 3.88 (s, 3H, OCH_3_), 7.30 (dd, J = 7.2, 4.8 Hz, 1H, H-5), 8.35 (m, 3H, H-2, H-4, H-6). ^13^C NMR (50 MHz, DMSO-d_6_) δ 31.41 (q), 50.91 (q), 103.75 (s), 117.87 (d), 118.23 (s), 128.95 (d), 136.38 (d), 143.73 (d), 147.56 (s), 163.99 (s). *Anal.* Calculated for C_10_H_10_N_2_O_2_ (MW: 190.20): C, 63.15; H, 5.30; N, 14.73%. Found: C, 63.28; H 5.15; N, 14.58%.

*Methyl-5-fluoro-1-methyl-1H-pyrrolo[2,3-b]pyridine-3- carboxylate* (**3b**). White solid; yield 90%; m.p. 128–129 °C; IR cm^−1^: 1738 (CO); ^1^H NMR (200 MHz, DMSO-d_6_) δ 3.83 (s, 3H, CH_3_), 3.88 (s, 3H, OCH_3_), 8.05 (dd, J = 9.2, 2.8 Hz, 1H, H-4), 8.39 (dd, J = 2.8, 1.8 Hz, 1H, H-6), 8.43 (s, 1H, H-2). ^13^C NMR (50 MHz, DMSO-d_6_) δ 31.73 (q), 51.02 (q), 103.76 (d, J*_C7a-F_* = 4.1 Hz), 114.48 (d, J*_C4-F_* = 21.8 Hz), 118.41 (d, J*_C3-F_* = 7.9 Hz), 132.19 (d, J*_C6-F_* = 29.4 Hz), 138.27 (d), 144.32 (s), 156.26 (d, J*_C5a-F_* = 242.4 Hz), 163.66 (s). *Anal.* Calculated for C_10_H_9_FN_2_O_2_ (MW: 208.19): C, 57.69; H, 4.36; N, 13.46%. Found: C, 57.53; H 4.15; N, 13.38%.

*Methyl-5-bromo-1-methyl-1H-pyrrolo[2,3-b]pyridine-3- carboxylate* (**3c**). White solid; yield 99%; m.p. 144–145 °C; IR cm^−1^: 1715 (CO); ^1^H NMR (200 MHz, DMSO-d_6_) δ 3.84 (s, 3H, CH_3_), 3.86 (s, 3H, CH_3_), 8.39 (d, J = 2.0 Hz, 2H, H-2, H-4), 8.45 (d, J = 2.0 Hz, 1H, H-6). ^13^C NMR (50 MHz, DMSO-d_6_) δ 31.64 (q), 51.10 (q), 103.45 (s), 113.36 (s), 119.73 (s), 130.63 (d), 137.85 (d), 143.88 (d), 146.03 (s) 163.57 (s). *Anal.* Calculated for C_10_H_9_BrN_2_O_2_ (MW: 269.09): C, 44.63; H, 3.37; N, 10.41%. Found: C, 44.87; H 3.51; N, 10.12%.

##### General Procedure for the Synthesis of 1-methyl-3-[3-(1-methyl-1*H*-indol-3-yl)-1,2,4-oxadiazol-5-yl]-1*H*-pyrrolo[2,3-b]pyridines (**1a**–**o**)

To a solution of the proper derivative **2** (2.6 mmol) in tetrahydrofuran (10 mL), sodium hydride (3.0 mmol, 79.3 mg) and molecular sieves were added. The reaction mixture was heated at 60 °C for 30 min. Then, the appropriate derivative **3** (1.3 mmol) was added and the mixture was heated under reflux for 4–6 h. The organic solvent was evaporated at reduced pressure and the resulting crude was purified by column chromatography using dichloromethane/ethyl acetate 1:1 as the eluent.

*1-Methyl-3-[3-(1-methyl-1H-indol-3-yl)-1,2,4-oxadiazol-5-yl]-1H-pyrrolo[2,3-b]pyridine* (**1a**). White solid; yield 75%; m.p. 203.2–203.9 °C;^1^H NMR (200 MHz, DMSO-d_6_) δ 3.94 (s, 3H, CH_3_), 3.98 (s, 3H, CH_3_), 7.24–7.36 (m, 2H, H-6″, H-7″), 7.40–7.46 (m, 1H, H-4″), 7.58–7.62 (m, 1H, H-5″), 8.12–8.16 (m, 1H, H-5), 8.24 (s, 1H, H-2″), 8.48–8.51 (m, 1H, H-6), 8.60 (dd, J = 2.1, 8.0 Hz, 1H, H-4), 8.65 (s, 1H, H-2). *Anal.* Calculated for C_19_H_15_N_5_O (MW: 329.36): C, 69.29; H, 4.59; N, 21.26%. Found: C, 69.55; H 4.51; N, 21.21%. HRMS: [MH]+, found 330.1351 C_19_H_15_N_5_O requires 330.1349.

*5-Bromo-1-methyl-3-[3-(1-methyl-1H-indol-3-yl)-1,2,4-oxadiazol-5-yl]-1H-pyrrolo[2,3-b]pyridine* (**1b**). White solid; yield 72% m.p. 230.7–231.9 °C; ^1^H NMR (200 MHz, DMSO-d_6_) δ 3.95 (s, 6H, 2xCH_3_), 7.31 (s, 2H, H-6″, H-7″), 7.59–7.63 (m, 1H, H-5″), 8.11–8.30 (m, 2H, H-2″, H-4″), 8.58–8.71 (m, 3H, H-2, H-4, H-6). *Anal.* Calculated for C_19_H_14_BrN_5_O (MW: 408.25): C, 55.90; H, 3.46; N, 17.15%. Found: C, 56.13; H 3.25; N, 16.84%. HRMS: [MH]+, found 408.0449 C_19_H_14_BrN_5_O requires 408.0454.

*5-Fluoro-1-methyl-3-[3-(1-methyl-1H-indol-3-yl)-1,2,4-oxadiazol-5-yl]-1H-pyrrolo[2,3-b]pyridine* (**1c**). White solid; yield 62 %; m.p. 227.4–228.4 °C; ^1^H NMR (200 MHz, DMSO-d_6_) δ 3.94 (s, 3H, CH_3_), 3.97 (s, 3H, CH_3_), 7.24–7.36 (m, 2H, H-6″, H-7″), 7.60 (d, J = 7.4 Hz, 1H, H-4″), 8.12 (d, J = 7.4 Hz, 1H, H-5″), 8.30 (s, 1H, H-2″), 8.38 (m, 1H, H-4) 8.51 (s, 1H, H-6), 8.74 (s, 1H, H-2). *Anal.* Calculated for C_19_H_14_FN_5_O (MW: 347.35): C, 65.70; H, 4.06; N, 20.16%. Found: C, 65.87; H 4.32; N, 20.44%. HRMS: [MH]+, found 348.1259 C_19_H_14_FN_5_O requires 348.1255.

*1-Methyl-3-[3-(5-bromo-1-methyl-1H-indol-3-yl)-1,2,4-oxadiazol-5-yl]-1H-pyrrolo[2,3-b]pyridine* (**1d**). Light brown solid; yield 75%; m.p. 248.2–248.9 °C; ^1^H NMR (200 MHz, DMSO-d_6_) δ 3.95 (s, 3H, CH_3_), 3.99 (s, 3H, CH_3_), 7.40–7.47 (m, 2H, H-4″, H-6″), 7.61 (d, J = 8.8 Hz, 1H, H-7″) 8.25 (s, 1H, H-4), 8.30 (s, 1H, H-2″), 8.50 (d, J = 6.2 Hz, 1H, H-5), 8.60 (d, J = 6.2 Hz, 1H, H-6), 8.68 (s, 1H, H-2). *Anal.* Calculated for C_19_H_14_BrN_5_O (MW: 408.25): C, 55.90; H, 3.46; N, 17.15%. Found: C, 56.08; H 3.31; N, 17.25%. HRMS: [MH]+, found 408.0456 C_19_H_14_BrN_5_O requires 408.0454.

*5-Bromo-3-[3-(5-bromo-1-methyl-1H-indol-3-yl)-1,2,4-oxadiazol-5-yl]-1-methyl-1H-pyrrolo[2,3-b]pyridine* (**1e**). White solid; yield 82%; m.p. 264.2–265.9 °C; ^1^H NMR (200 MHz, DMSO-d_6_) δ 3.94 (s, 3H, CH_3_), 3.96 (s, 3H, CH_3_), 7.46 (d, J = 8.1 Hz, 1H, H-6″) 7.61 (d, J = 8.1 Hz, 1H, H-7″), 8.25 (s, 1H, H-4″), 8.36 (s, 1H, H-2″), 8.58 (s, 1H, H-2), 8.72 (m, 2H, H-4, H-6). *Anal.* Calculated for C_19_H_13_Br_2_N_5_O (MW: 487.15): C, 46.84; H, 2.69; N, 14.38%. Found: C, 46.97; H 3.00; N, 14.12%. HRMS: [MH]+, found 485.9561 C_19_H_13_Br_2_N_5_O requires 485.9559.

*5-Fluoro-3-[3-(5-bromo-1-methyl-1H-indol-3-yl)-1,2,4-oxadiazol-5-yl]-1-methyl-1H-pyrrolo[2,3-b]pyridine* (**1f)**. White solid; yield 75%; m.p. 255.3–256.2 °C; ^1^H NMR (200 MHz, DMSO-d_6_) δ 3.94 (s, 3H, CH_3_), 3.97 (s, 3H, CH_3_), 7.46 (d, J = 8.7 Hz, 1H, H-6″), 7.61 (d, J = 8.7 Hz, 1H, H-7″), 8.23 (s, 1H, H-2″), 8.35–8.37 (m, 2H, H-4″, H-4), 8.51 (s, 1H, H-6), 8.76 (s, 1H, H-2). *Anal.* Calculated for C_19_H_13_BrFN_5_O (MW: 426.24): C, 53.54; H, 3.07; N, 16.43%. Found: C, 53.68; H 2.98; N, 16.71%. HRMS: [MH]+, found 426.0358 C_19_H_13_BrFN_5_O requires 426.0360.

*3-[3-(5-Fluoro-1-methyl-1H-indol-3-yl)-1,2,4-oxadiazol-5-yl]-1-methyl-1H-pyrrolo[2,3-b]pyridine* (**1g**). Light brown solid; yield 72%; m.p. 236.0–236.7 °C; ^1^H NMR (200 MHz, DMSO-d_6_) δ 3.95 (s, 3H, CH_3_), 3.99 (s, 3H, CH_3_), 7.19 (t, J = 8.8 Hz, 1H, H-6″), 7.41–7.48 (m, 1H, H-7″), 7.63 (m, 1H, H-4″), 7.80 (d, J = 8.0 Hz, 1H, H-4), 8.33 (s, 1H, H-2″), 8.51 (d, J = 3.4 Hz, 1H, H-6), 8.60 (d, J = 8.0 Hz, 1H, H-5), 8.68 (s, 1H, H-2). *Anal.* Calculated for C_19_H_14_FN_5_O (MW: 347.35): C, 65.70; H, 4.06; N, 20.16%. Found: C, 65.55; H 4.31; N, 20.25%. HRMS: [MH]+, found 348.1253 C_19_H_14_FN_5_O requires 348.1255

*5-Bromo-3-[3-(5-fluoro-1-methyl-1H-indol-3-yl)-1,2,4-oxadiazol-5-yl]-1-methyl-1H-pyrrolo[2,3-b]pyridine* (**1h**). Light brown solid; yield 60%; m.p. 255.2–256.8 °C; ^1^H NMR (200 MHz, DMSO-d_6_) δ 3.95 (s, 3H, CH_3_), 3.96 (s, 3H, CH_3_), 7.19 (t, J = 8.3 Hz, 1H, H-7″), 7.64 (m, 1H, H-4″), 7.77 (d, J = 8.3, 1.7 Hz, 1H, H-6″), 8.37 (s, 1H, H-2″), 8.58 (s, 1H, H-2), 8.72 (m, 2H, H-4, H-6). *Anal.* Calculated for C_19_H_13_BrFN_5_O (MW: 426.24): C, 53.54; H, 3.07; N, 16.43%. Found: C, 53.28; H 3.15; N, 16.26%. %. HRMS: [MH]+, found 426.0363 C_19_H_13_BrFN_5_O requires 426.0360

*5-Fluoro-3-[3-(5-fluoro-1-methyl-1H-indol-3-yl)-1,2,4-oxadiazol-5-yl]-1-methyl-1H-pyrrolo[2,3-b]pyridine* (**1i**). White solid; yield 55%; m.p. 265.2–266.8 °C; ^1^H NMR (200 MHz, DMSO-d_6_) δ 3.95 (s, 3H, CH_3_), 3.98 (s, 3H, CH_3_), 7.20 (td, J = 2.6, 9.0 Hz, 1H, H-6″), 7.61–7.68 (m, 1H, H-4″), 7.79 (d, J = 2.6 Hz, 1H, H-6″), 8.35–8.41 (m, 2H, H-2″, H-4), 8.51–8.53 (m, 1H, H-6), 8.76 (s, 1H, H-2). *Anal.* Calculated for C_19_H_13_F_2_N_5_O (MW: 365.34): C, 62.46; H, 3.59; N, 19.17%. Found: C, 62.68; H 3.38; N, 19.42%. HRMS: [MH]+, found 366.1157 C_19_H_13_F_2_N_5_O requires 366.1160

*3-[3-(5-Chloro-1-methyl-1H-indol-3-yl)-1,2,4-oxadiazol-5-yl]-1-methyl-1H-pyrrolo[2,3-b]pyridine* (**1j**). White solid; yield 65%; m.p. 242.8–243.7 °C; ^1^H NMR (200 MHz, DMSO-d_6_) δ 3.94 (s, 3H, CH_3_), 3.97 (s, 3H, CH_3_), 7.51–7.20 (m, 1H, H-6″), 7.64 (d, J = 9.3 Hz, 1H, H-7″),8.65 (s, 1H, H-4″), 8.57 (d, J = 7.1 Hz, 1H, H-5), 8.49 (d, J = 2.5 Hz, 1H, H-4), 8.30 (s, 1H, H-2″), 8.09 (s, 1H, H-2), 7 3.95 (d, J = 7.1 Hz, 1H, H-6). *Anal.* Calculated for C_19_H_14_ClN_5_O (MW: 363.80): C, 62.73; H, 3.88; N, 19.25%. Found: C, 62.58; H 3.71; N, 19.37%. HRMS: [MH]+, found 364.0962 C_19_H_14_ClN_5_O requires 364.0960.

*5-Bromo-3-[3-(5-chloro-1-methyl-1H-indol-3-yl)-1,2,4-oxadiazol-5-yl]-1-methyl-1H-pyrrolo[2,3-b]pyridine* (**1k**). White solid; yield 75%; m.p. 270.2–271.3 °C; ^1^H NMR (200 MHz, DMSO-d_6_) δ 3.95 (s, 3H, CH_3_), 3.97 (s, 3H, CH_3_), 7.35 (dd, J = 8.7, 2.0 Hz, 1H, H-6″), 7.66 (d, J = 8.7 Hz, 1H, H-7″), 8.09 (d, J = 2.0 Hz, 1H, H-4″), 8.38 (s, 1H, H-2″), 8.58 (d, J = 2.2 Hz, 1H, H-4), 8.70 (d, J = 2.2 Hz, 1H, H-6), 8.74 (s, 1H, H-2). *Anal.* Calculated for C_19_H_13_BrClN_5_O (MW: 442.70): C, 51.55; H, 2.96; N, 15.82%. Found: C, 51.28; H 3.11; N, 16.02%. HRMS: [MH]+, found 442.0068 C_19_H_13_BrClN_5_O requires 442.0065.

*5-Fluoro-3-[3-(5-chloro-1-methyl-1H-indol-3-yl)-1,2,4-oxadiazol-5-yl]-1-methyl-1H-pyrrolo[2,3-b]pyridine* (**1l**). White solid; yield 85%; m.p. 258–259 °C; ^1^H NMR (200 MHz, DMSO-d_6_) δ 3.94 (s, 3H, CH_3_), 3.97 (s, 3H, CH_3_), 7.34 (d, J = 8.3 Hz, 1H, H-6″), 8.07 (m, 1H, H-7″), 7.64 (m, 1H, H-4″), 8.39 (m, 2H, H-6, H-4’), 8.52 (s, 1H, H2″), 8.77 (s, 1H, H-2). *Anal.* Calculated for C_19_H_13_ClFN_5_O (MW: 381.79): C, 59.77; H, 3.43; N, 18.34%. Found: C, 59.68; H 3.25; N, 18.56%. HRMS: [MH]+, found 382.0867 C_19_H_13_ClFN_5_O requires 382.0865.

*3-[3-(5-Methoxy-1-methyl-1H-indol-3-yl)-1,2,4-oxadiazol-5-yl]-1-methyl-1H-pyrrolo[2,3-b]pyridine* (**1m**). White solid; yield 70%; m.p. 197–198 °C; ^1^H NMR (200 MHz, DMSO-d_6_) δ 3.86 (s, 3H, CH_3_), 3.90 (s, 3H, CH_3_), 3.98 (s, 3H, OCH_3_), 6.96 (d, J = 9.0 Hz, 1H, H-6″), 7.61 (s, 1H, H-2″), 7.55–7.32 (m, 2H, H-7″, H-4″), 8.17 (s, 1H, H-2), 8.49 (d, J = 4.0 Hz, 1H, H-6), 8.60 (m, 2H, H-4, H-5). *Anal.* Calculated for C_20_H_17_N_5_O_2_ (MW: 359.38): C, 66.84; H, 4.77; N, 19.49%. Found: C, 66.59; H 4.89; N, 19.23%. HRMS: [MH]+, found 360.1456 C_20_H_17_N_5_O_2_ requires 360.1455.

*5-Bromo-3-[3-(5-methoxy-1-methyl-1H-indol-3-yl)-1,2,4-oxadiazol-5-yl]-1-methyl-1H-pyrrolo[2,3-b]pyridine* (**1n**). White solid; yield 65%; m.p. 229.6–230.2 °C; ^1^H NMR (200 MHz, DMSO-d_6_) δ 3.88 (s, 3H, CH_3_). 3.90 (s, 3H, CH_3_), 3.97 (s, 3H, OCH_3_), 6.95 (d, J = 8.3 Hz, 1H, H-6″), 7.50 (d, J = 8.3 Hz, 1H, H-7″), 7.62 (d, J = 1.3 Hz, 1H, H-4″), 8.21 (s, 1H, H-2″), 8.58 (s, 1H, H-2), 8.71 (m, 2H, H-4, H-6). *Anal.* Calculated for C_20_H_16_BrN_5_O_2_ (MW: 438.28): C, 54.81; H, 3.68; N, 15.98%. Found: C, 54.78; H 3.95; N, 15.78%. HRMS: [MH]+, found 438.0562 C_20_H_16_BrN_5_O_2_ requires 438.0560.

*5-Fluoro-3-[3-(5-methoxy-1-methyl-1H-indol-3-yl)-1,2,4-oxadiazol-5-yl]-1-methyl-1H-pyrrolo[2,3-b]pyridine* (1o). White solid; yield 80%; m.p. 218–219 °C ^1^H NMR (200 MHz, DMSO-d_6_) δ 3.78 (s, 3H, CH_3_), 3.90 (s, 3H, CH_3_), 3.96 (s, 3H, OCH_3_), 6.95 (dd, J = 8.9, 2.4 Hz, 1H, H-6″), 7.54 (m, 2H, H-7″, H-4″), 8.22 (m, 1H, H-4), 8.50 (s, 1H, H-2″), 8.37 (dd, J = 8.9, 2.7 Hz, 1H, H-6), 8.72 (s, 1H, H-2). *Anal.* Calculated for C_20_H_16_FN_5_O_2_ (MW: 377.37): C, 63.65; H, 4.27; N, 18.56%. Found: C, 63.78; H 4.10; N, 18.63%. HRMS: [MH]+, found 378.1359 C_20_H_16_FN_5_O_2_ requires 378.1361.

### 3.2. Biology

Compounds **1a**–**o**, prepared as described above, were dissolved in dimethyl sulfoxide (DMSO) and then diluted in culture medium to have a DMSO concentration not exceeding 0.1%. MCF-7 (human breast cancer), HeLa (cervix adenocarcinoma), HCT-116 (human colorectal carcinoma), and CaCo2 (colorectal carcinoma) cell lines were obtained from American Type Culture Collection, Rockville, MD, USA and grown in Dulbecco’s modified Eagle’s medium (DMEM) supplemented with 10% fetal, 10% fetal bovine serum (FBS), penicillin (100 U/mL), streptomycin (100 μg/mL), and gentamicin (5 μg/mL). All the cells were maintained in log phase by seeding twice a week at a density of 3 × 10^8^ cells/L in a humidified 5% CO_2_ atmosphere, at 37 °C. In all experiments, cells were left to incubate overnight to allow adhesion before treatment with the compounds or vehicle alone (control cells). In selected experiments CaCo2 cells were treated 15 days after confluence, at which time the cells are differentiated in normal intestinal-like cells [43].

No differences were found between cells treated with DMSO 0.1% and untreated cells in terms of cell number and viability.

#### 3.2.1. Viability Assay in Vitro

Cytotoxic activity of compounds **1k** and **1n** against human tumor cell lines (MCF-7, HeLa, Caco-2, and HCT-116) and intestinal-like differentiated cells was determined by the MTT colorimetric assay commonly used to study inhibition of cellular proliferation. Briefly, cells were seeded at 2 × 10^4^ cells/well in 96-well plates containing 200 μL DMEM. When appropriated, cells were treated with vehicle alone (0.1% DMSO, control) or various concentrations (0.01–100 μM) of the drugs in DMEM and let them incubate for 72 h. Then cells were washed, and 50 μL of FBS-free medium containing 5 mg/mL MTT was added. The medium was discarded after 2 h incubation at 37 °C by centrifugation, and formazan blue formed in the cells was dissolved in DMSO. The absorbance, measured at 570 nm on a microplate reader (Bio-RAD, Hercules, CA, USA), of MTT formazan of control cells were taken as 100% of viability.

Cytotoxicity of the compounds was defined as the IC_50_ value which represents the molar concentration of the compound that inhibits 50% cell viability. IC_50_ values were calculated by the dose-response inhibition model using GraphPad Prism 5.02 from GraphPad Software (San Diego, CA, USA). Each experiment was repeated at least three times in triplicate to obtain the mean values.

#### 3.2.2. Cell Cycle Analysis

MCF-7 cells (5.0 × 10^4^ cells/cm^2^) were seeded in triplicate in 24-wells culture plates. After an overnight incubation, the cells were washed with fresh medium and incubated with the compounds or vehicle alone (control cells) in DMEM for 24 h. After trypsinization, aliquots of 1.0 × 10^6^ cells were washed with PBS and incubated in the dark in PBS containing 20 μg/mL propidium iodide (PI) and 200 μg/mL RNase, for 30 min, at room temperature. Then samples of at least 1.0 × 10^4^ cells were subjected to fluorescence-activated cell sorting (FACS) analysis by Epics XL™ flow cytometer using Expo32 software (Beckman Coulter, Fullerton, CA, USA).

#### 3.2.3. Cell Death Detection

Cell death was assessed by staining the cells with AnnexinV-FITC and propidium iodide (PI) (Sigma-Aldrich, Steinheim, Germany) according to the manufacturer’s instructions (eBioscience, San Diego, CA, USA) and subsequent analysis by flow cytometry. MCF-7 cells (5.0 × 104 cells/cm^2^) were seeded in triplicate in 24-wells culture and after an overnight incubation, washed with fresh medium and incubated for 24 h with the compounds or vehicle alone (control cells) in DMEM. After trypsinization, 1.0 × 10^6^ cells/mL in combining buffer were incubated with Annexin V-FITC and PI solution at room temperature in the dark for 15 min. Then samples of at least 1.0 × 10^4^ cells were subjected to FACS analysis using appropriate 2-bidimensional gating method.

Acridine orange/ethidium bromide (AO/EB) fluorescence staining was also used to identify cell apoptosis. After 24 h treatment, the cells were washed with PBS and stained with AO/EB solution (100 μg/mL, 1:1). After 20 s the AO/EB solution was discarded and cells were immediately visualized by fluorescence microscopy (Leica; Wetzlar, Germany). Multiple photos were taken at randomly-selected areas of the well to obtain representative data.

## 4. Conclusions

New analogs of nortopsentin, in which the central imidazole ring of the natural lead was replaced by a 1,2,4-oxadiazole moiety and in which a 7-azaindole portion substituted the original indole moiety, were efficiently synthesized. Among the synthesized derivatives, **1k** and **1n** showed the best cytotoxic activity against several human cancer cell lines. The mechanism of the anti-proliferative effect on MCF-7 was pro-apoptotic, being associated with externalization of plasma membrane phosphatidylserine, chromatin condensation, and membrane blebbing. Finally, the compounds induced an accumulation of cells in G0–G1 phase suggesting that they can affect the cell mechanisms promoting the DNA duplication.

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
