# Peer review of "New 1,2,4-Oxadiazole Nortopsentin Derivatives with Cytotoxic Activity"

_marinedrugs, 2019, doi:10.3390/md17010035_

Round 1

Reviewer 1 Report

This manuscript presents the synthesis and bioactivity of nortopsentin derivatives. The work seems carefully done and the introduction, experiment design, analyses, results, and conclusions are clearly presented. The term ‘anticancer’ should be changed to ‘cytotoxic’ according to the NIH guidelines for cancer drug discovery studies, because cell lines were used for bioassay.

The manuscript needs corrections as follows.

(1) Line 91: The atom-numbering should be given to the structural formula of compounds 1a-1o.

(2) Lines 221-380: ‘DMSO’ should be corrected to DMSO-d6’.

(3) Lines 290-383: The authors should provide MS data for confirmation of the final synthetic compounds 1a-1o.

(4) Line 294: A chemical shift ‘.65’ should be corrected.

(5) Line 350: A chemical shift ‘7 3.95’ should be corrected.

Author Response

Response to Referee #1

This manuscript presents the synthesis and bioactivity of nortopsentin derivatives. The work seems carefully done and the introduction, experiment design, analyses, results, and conclusions are clearly presented. The term ‘anticancer’ should be changed to ‘cytotoxic’ according to the NIH guidelines for cancer drug discovery studies, because cell lines were used for bioassay.

ü We changed the term anticancer with cytotoxic according to referee suggestion.

The manuscript needs corrections as follows.

(1) Line 91: The atom-numbering should be given to the structural formula of compounds 1a-1o.

(2) Lines 221-380: ‘DMSO’ should be corrected to ‘DMSO-d6’.

(3) Lines 290-383: The authors should provide MS data for confirmation of the final synthetic compounds 1a-1o.

(4) Line 294: A chemical shift ‘.65’ should be corrected.

(5) Line 350: A chemical shift ‘7 3.95’ should be corrected.

ü We made all corrections as suggested at points (2), (4) and (5).

ü We gave atom-numbering to the structural formula of compounds 1a-1o.

ü We provided MS data for confirmation of the final synthetic compounds 1a-1o.

Reviewer 2 Report

In this manuscript, Casioferro et al. reported the anticancer activity of a new 1,2,4-oxadiazole nortopsentin derivatives in HCT-116 colon rectal carcinoma cell lines. Though this manuscript seems a good piece of work, there are few conceptual advancements needed for this manuscript. The authors need to explain all the data clearly with relevant rationale and should be discussed more relevantly using updated literature. Also, the authors need to perform more in vitro and some of the in vivo experiments to validate the anticancer activity of these synthesized chemical derivatives. By merely measuring the cytotoxicity of these derivatives in the cancer cell line never considered as an anticancer activity. Also, the authors need to perform fluorescence staining for live and dead cells to claim their cytotoxicity property of these chemical derivatives. These experiments are mandatory. Also, the authors need to use in vivo cancer model to validate their proposed hypothesis. Also, it is essential that the authors need to demonstrate the molecular signaling by which these derived chemicals exhibit anticancer activity. Even, if the authors propose some molecular signaling pathway while revising the manuscript, the authors should include both pharmacological as well as gene silencing approaches for unraveling the proposed molecular signaling pathway. Also, the authors need to revise the manuscript carefully for the typos and Grammatical mistakes. Overall, this manuscript needs extensive major revision.

Author Response

Response to Referee #2

In this manuscript, Casioferro et al. reported the anticancer activity of a new 1,2,4-oxadiazole nortopsentin derivatives in HCT-116 colon rectal carcinoma cell lines. Though this manuscript seems a good piece of work, there are few conceptual advancements needed for this manuscript. The authors need to explain all the data clearly with relevant rationale and should be discussed more relevantly using updated literature.

ü We added some comments according to referee suggestion in Results and Discussion.

Also, the authors need to perform more in vitro and some of the in vivo experiments to validate the anticancer activity of these synthesized chemical derivatives. By merely measuring the cytotoxicity of these derivatives in the cancer cell line never considered as an anticancer activity.

ü We changed the term anticancer with cytotoxic according to referee 2 suggestion.

Also, the authors need to perform fluorescence staining for live and dead cells to claim their cytotoxicity property of these chemical derivatives. These experiments are mandatory.

ü We performed fluorescence staining for live and dead cells commenting the obtained data in the text. Moreover, we inserted the related figure 4 and updated the “Materials and Methods” paragraph accordingly.

Also, the authors need to use in vivo cancer model to validate their proposed hypothesis. Also, it is essential that the authors need to demonstrate the molecular signaling by which these derived chemicals exhibit anticancer activity. Even, if the authors propose some molecular signaling pathway while revising the manuscript, the authors should include both pharmacological as well as gene silencing approaches for unraveling the proposed molecular signaling pathway.

ü We thank the referee for suggestions. In this work, however, the investigation on the synthesized compounds was limited to cell cultures to study their cytotoxic activity, according to referee 2 suggestion.

Also, the authors need to revise the manuscript carefully for the typos and Grammatical mistakes. Overall, this manuscript needs extensive major revision.

ü We revised the manuscript with care for typos and grammatical errors.

Round 2

Reviewer 2 Report

Though the authors were provided with a list of comments to improve the scientific merit of the manuscript, the authors made very minimal efforts by merely performing language tweaking without performing the recommended mechanistic signaling approach in this revised manuscript. These experiments are most important for the proposed claim. However, now the author rephrased the anticancer activity into cytotoxic activity, which further weakens the innovation and novelty of the work. Hence, this manuscript needs to be revised significantly.

Author Response

Dear Editor,

Upon the comments and suggestions of the reviewers, we have improved the quality of the manuscript by making all the changes requested by the reviewer 1 and partially those requested by the reviewer 2.

After the revision of the manuscript the reviewer 1 did not raise further comments, whereas, reviewer 2 continued to require mechanistic studies on our compounds.

We believe that the biological data we have reported are sufficient to define our compounds as cytotoxic agents.

In addition, the biological tests we have performed are perfectly in line with those requested by the journal Marine Drugs in which we have been publishing for several years (see references 1-5). Such articles attracted interests of researchers in Drug Discovery, reaching citation benchmarking of 86-96 percentile compared with the average of similar documents, with the exception of ref 1, just published.

Moreover, in none of the articles published in the last 5 volumes (see for instance references 6-11), in which is reported the synthesis and biological activity of new derivatives, in vivo tests are reported and only in one of those articles (Ref. 8) are present more detailed studies to identify the mechanism of action of the reported compounds with no citation.

Thus, we ask the Academic Editor of Marine Drugs for evaluating the possibility to accept this manuscript.

Looking forward to hearing from you

Dr. Barbara Parrino

References

1. Carbone A. et al. Synthesis and antiproliferative activity of 2,5-bis(3′-indolyl)pyrroles, analogues of the marine alkaloid nortopsentin. Mar. Drugs 2013, 11, 643−654.

2. Carbone A. et al. Synthesis and antiproliferative activity of thiazolyl-bis-pyrrolo[2,3-b]pyridines and indolyl-thiazolyl-pyrrolo[2,3-c]pyridines, nortopsentin analogues. Mar. Drugs 2015, 13, 460–492.

3. Parrino B. et al. 3-[4-(1H-Indol-3-yl)-1,3-thiazol-2-yl]-1H-pyrrolo[2,3-b]pyridines, nortopsentin analogues with antiproliferative activity. Mar. Drugs 2015, 13, 1901–1924.

4. Spanò V. et al. Synthesis and antitumor activity of new thiazole nortopsentin analogs. Mar. Drugs 2016, 14, 226–243.

5. Carbone A. et al. New thiazole nortopsentin analogues inhibit bacterial biofilm formation. Mar. Drugs 2018, 16, 274.

6. Huang Y. et al. Synthesis and Evaluation of Some New Aza-B-homocholestane Derivatives as Anticancer Agents. Mar. Drugs 2014, 12, 1715-1731;

7. Cui J. et al. Synthesis and in Vitro Antiproliferative Evaluation of Some B-norcholesteryl Benzimidazole and Benzothiazole Derivatives. Mar. Drugs 2015, 13, 2488-2504;

8. Fuwa H. et al. A Synthetic Analogue of Neopeltolide, 8,9-Dehydroneopeltolide, Is a Potent Anti-Austerity Agent against Starved Tumor Cells. Mar. Drugs 2017, 15, 320

9. Zhang J. et al. Synthesis of Novel Chitin Derivatives Bearing Amino Groups and Evaluation of Their Antifungal Activity. Mar. Drugs 2018, 16, 380;

10. Bhat C. et al. Synthesis and Antiproliferative Activity of Marine Bromotyrosine Purpurealidin I and Its Derivatives. Mar. Drugs 2018, 16, 481ù

11. Matsubara T. et al. Asymmetric Synthesis and Cytotoxicity Evaluation of Right-Half Models of Antitumor Renieramycin Marine Natural Products. Mar. Drugs 2019, 17, 3;